# Six groups of ground-dwelling arthropods show different diversity responses along elevational gradients in the Swiss Alps

José D. Gilgado[1,2], Hans-Peter Rusterholz[1], Brigitte Braschler[1], Stephan Zimmermann[3]*, Yannick Chittaro[4], Bruno Baur[1]

**1** Section of Conservation Biology, Department of Environmental Sciences, University of Basel, Basel, Switzerland, **2** Grupo de Investigación de Biología del Suelo y de los Ecosistemas Subterráneos, Departamento de Ciencias de la Vida, Facultad de Ciencias, Universidad de Alcalá, Alcalá de Henares, Madrid, Spain, **3** Forest Soils and Biogeochemistry, Swiss Federal Institute for Forest, Snow and Landscape Research (WSL), Birmensdorf, Zürich, Switzerland, **4** Info Fauna, Centre Suisse de Cartographie de la Faune (CSCF), Neuchâtel, Switzerland

* stefan.zimmermann@wsl.ch

**Data Availability Statement:** All relevant data are within the manuscript and its Supporting Information files.

## Abstract

Elevational gradients along mountain slopes offer opportunities to study key factors shaping species diversity patterns. Several environmental factors change over short distances along the elevational gradient in predictable ways. However, different taxa respond to these factors differently resulting in various proposed models for biodiversity patterns along elevational transects. Using a multi-taxa approach, we investigated the effects of elevation, area, habitat and soil characteristics on species richness, individual abundance and species composition of six groups of ground-dwelling arthropods along four transect lines in the Swiss National Park and its surroundings (Eastern Alps). Spiders, millipedes, centipedes, ants, ground beetles and rove beetles were sampled using standardized methods (pitfall traps, cardboard traps, visual search) in 65 sites spanning an elevational range from 1800 to 2750 m a.s.l.. A total of 14,782 individuals comprising 248 species were collected (86 spider, 74 rove beetle, 34 ground beetle, 21 millipede, 19 centipede and 14 ant species). Linear mixed model-analysis revealed that rarefied species richness in five out of the six arthropod groups was affected by elevation (the quadratic term of elevation provided the best fit in most cases). We found three different patterns (linear decrease in centipedes, low elevation plateau followed by a decrease in ants and rove beetles, and midpoint peak in spiders and millipedes). These patterns were only partially mirrored when considering individual abundance. Elevation influenced species composition in all groups examined. Overall, elevation was the most important factor explaining the diversity patterns, while most local habitat and soil characteristics have little influence on these patterns. Our study supports the importance of using multi-taxa approaches when examining effects of elevational gradients. Considering only a single group may result in misleading findings for overall biodiversity.

**Funding:** The Swiss National Park Direction and Research Commission provided logistical help and funding (JDG, BBa; Grants 2019-12, 2020-08). Additional funding was obtained from the 'Stiftung Sammlung Naturmuseum Chur' (JDG, BBa; Grant 07/20). The funders had no role in study design, data collection and analysis, decision to publish, or preparation of the manuscript.

**Competing interests:** The authors have declared that no competing interests exist.

## Introduction

Most mountain areas harbour a higher biodiversity than the surrounding lowlands [1–4]. This can be explained–among others–by the highly diverse topography of the mountains, their variety of microhabitats and heterogeneous microclimatic conditions, the mountains' role of retaining relict populations during periods of glaciation, and the fact that high mountain areas are usually less impacted by human activities than lowland areas [1, 2, 5]. Furthermore, many rare and endemic species, which are often poor dispersers occurring in isolated populations, are restricted to mountain areas [5–7].

Biodiversity is not evenly distributed along mountain slopes. Habitats change along the elevational gradient in a predictable way, best known as vegetation belts in plant communities [8]. However, other biodiversity patterns along elevational gradients are less clear or not consistent. While species richness is usually lowest in the highest part of the mountains, different patterns within and among taxonomical groups have been reported along the elevational gradient [9, 10]. McCain & Grytnes [9] described four main patterns: (1) "Decreasing species richness", with a relatively constant reduction in number of species towards higher elevations; (2) "Low plateau species richness", with a constant number of species richness in the lower part of the elevational gradient, and a constant reduction higher up; (3) "Low-elevation plateau with a mid-peak", with a high richness across low elevations and a peak recorded more than 300 m from the valley bottom; and (4) "Midpeak", with a peak in the number of species at medium elevation (around 25% higher than in the lowland and on the mountain top). Deviations from these four common patterns were only reported in a few cases, e.g. in salamanders and lichens, in which species richness increased with elevation [9, 11].

Different, mutually not exclusive, explanations have been proposed for the four observed common patterns. The elevational gradient is frequently considered as a proxy for intercorrelated variables such as the steady temperature decrease with increasing elevation, the decreasing air pressure, the increasing solar radiation and the decreasing length of the vegetation period at higher elevation, while other environmental variables such as precipitation and soil quality also change along elevation but do so in different ways depending on the geographic region [9, 12, 13]. McCain & Grytnes [9] suggested that the various explanations for elevational patterns of species richness can be grouped into four coarse categories: Firstly, "climate", which includes changes in temperature, precipitation, length of the vegetation period and other variables along the elevational gradient. Secondly, "space", which regards the species-area relationship and the fact that elevational bands have usually less area in higher parts of the gradient, as well as the spatial constraint hypothesis (mid-domain effect; MDE). Thirdly, "evolutionary history", which relates to the speciation rate and extinction rates not being constant along the elevational gradient leading to a single diversity optimum, as well as niche conservatism based on the fact that most modern taxonomical groups evolved in tropical-like conditions. Fourthly, "biotic processes", which include competition, ecotone effects, the heterogeneity of habitats, etc. In all four categories, there are processes that lead to an increase or to a decrease of species richness at a given site along an elevational gradient. However, not all factors can be assigned exclusively to one of these four categories listed above. For example, Antonelli et al. [14] showed that species richness of terrestrial tetrapods along elevational gradients correlates with erosion rates and heterogeneity of soil types. Soil properties also influence ground-dwelling invertebrate diversity locally (e. g. [15–18]), but soil properties also vary along elevational gradients [19]. Furthermore, intense human activities in the lowlands and lower parts of mountain slopes may artificially reduce local species richness, resulting in a "midpeak" pattern [20, 21]. Moreover, sampling designs, in which individuals were not collected over the entire elevation gradient, could result in truncated patterns [9].

Available land area is generally considered as a main driver of species diversity on mountain slopes [22–25]. Due to the conical shape of many mountains, land area decreases with increasing elevation, thereby narrowing opportunities for life [22, 26]. Furthermore, patterns of species richness could be shaped by geometric constraints (e.g. the mid-domain effect; MDE; [27, 28]). MDE arises from the fact that the random distribution of ranges within a restricted geographical domain (e.g. the lower and upper elevational limit of a mountain slope) will always show the greatest degree of overlap in the mid-range (i.e. at mid-elevation), without the need for any climatic or environmental factors for the placement of the ranges [29].

Biodiversity studies along elevational gradients have previously predominantly focused on single taxonomical groups. More recently, however, an increased number of studies using a multi-taxa approach have been published (e.g. [5, 10, 30–32]). The main objective of our study is to unravel factors shaping patterns of species richness, abundance and species composition in spiders, millipedes, centipedes, ants, ground beetles and rove beetles along elevational gradients in four valleys in the Swiss National Park (SNP) and adjacent areas. The SNP is a strict nature reserve in the Eastern Alps with no human disturbance or land-use change since 1914 (IUCN-category Ia). Thus, the elevational distribution of these arthropod species within the SNP is not confounded by human activities in the past century. The ground-dwelling arthropods investigated in our study play a major role in soil dynamics and nutrient recycling [33]. Moreover, most species of the six arthropod groups considered are poor dispersers during at least a part of their life cycle, which suggests that they will be strongly affected by local environmental conditions.

We explored the relationships between species richness, abundance and species composition of the six arthropod groups and elevation, land area, habitat type and various site and soil characteristics at the level of elevational bands. We considered several alternative models that included environmental and elevational aspects, geometric constraints (mid-domain effect; MDE) and land area to explain patterns of diversity in the six arthropod groups. We focused on the following questions:

1. Do the six groups of ground-dwelling arthropods show similar patterns of species-richness change along the same elevational gradient, and can these patterns be assigned to one of those suggested by McCain and Grytnes [9]?

2. Do relationships between individual abundance and elevation mirror the patterns observed for species richness for the corresponding arthropod groups?

3. Which environmental variables (elevation, available area, soil characteristics, substrate and vegetation type) play a major role in explaining patterns of species richness, abundance and species composition along mountain slopes for the six arthropod groups.

## Material and methods

### Study area

The SNP was established in 1914 in the Eastern Alps, Switzerland (46°39'N, 10°12' E). As a strict nature reserve (category Ia; [34]), its main targets are (1) ecosystem protection without any influence of humans or domestic animals, and (2) scientific research. There is no habitat management, and public access is permitted only on marked paths in summer months. The SNP measures 170.3 km$^2$ and includes an elevational range from 1315 m to 3173 m a.s.l. ([35]; Fig 1). Forests cover 32% of the park area with a tree line around 2200 m a.s.l., alpine grasslands 20%, waters 1%, and rocks and scree slopes 47% [35]. We established elevational transect lines in four valleys. The geology of the two valleys Val dal Botsch and Val Trupchun/Val

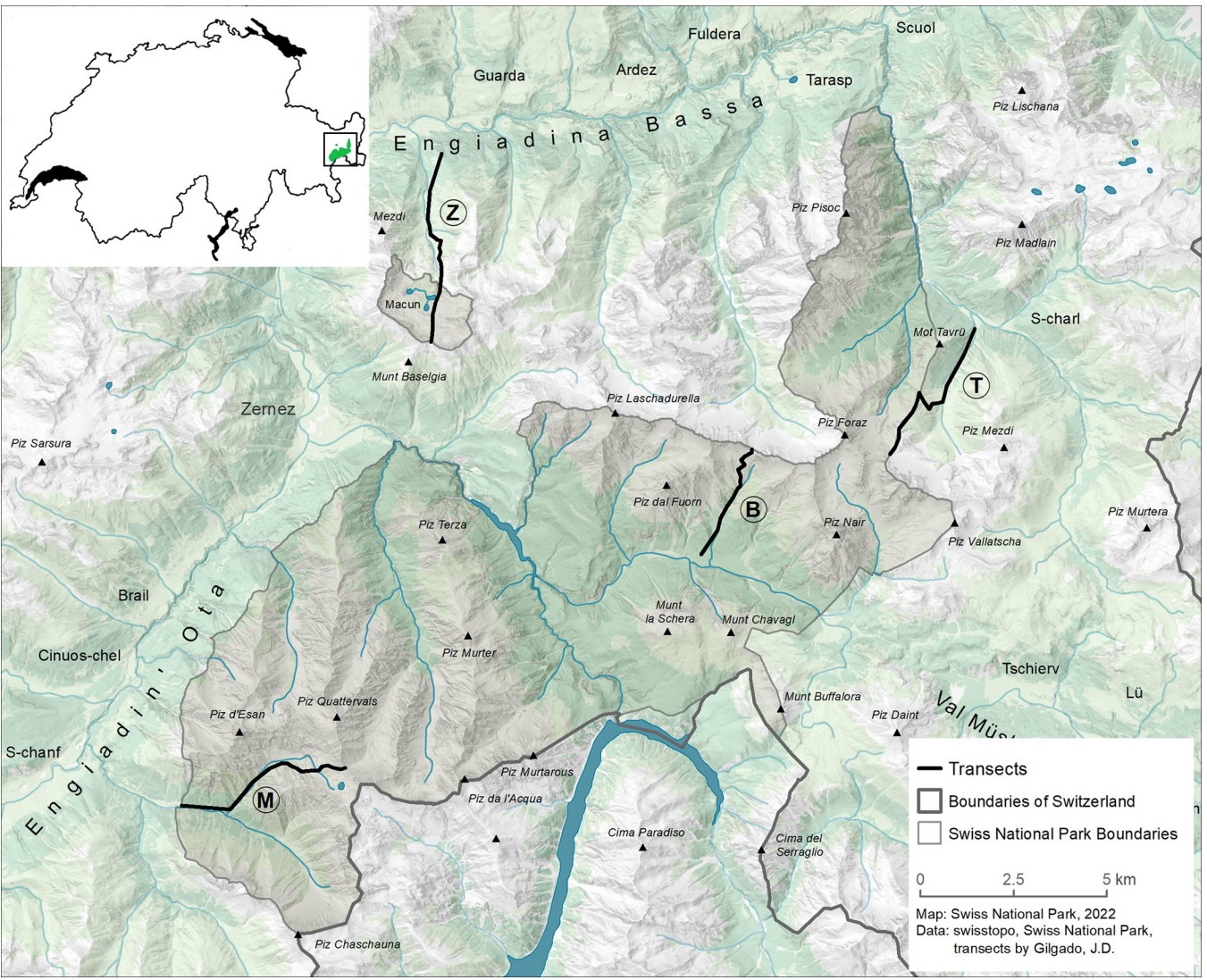

**Fig 1. Map of the four transect lines examined in the Swiss National Park and its surroundings.** Letters indicate the names of the transects: B – Val dal Botsch; M–Val Trupchun/Val Müschauns; T–Val Tavrü, and Z–Val Zeznina/Macun. Reprinted from a map made by T. Estermann for the Swiss National Park under a CC BY license, with permission from the Swiss National Park Direction, original copyright Swiss National Park 2022. Data swisstopo, Swiss National Park, transects by J. D. Gilgado.

Müschauns is largely characterized by carbonate rocks, especially dolomite and marl limestone [36]. Both valleys are entirely situated within the boundaries of the national park. Adjacent areas to the SNP are covered by forest, rocks and scree slopes and grassland, which is extensively managed by livestock farming (cattle, sheep). The lower part of Val Tavrü is characterized by moraine, which is replaced towards higher elevations by siltstone and conglomerates and over 2300 m a.s.l. by dolomite. In the Val Zeznina north of the SNP, the lower part is also dominated by moraine, while the upper part of the valley consists of amphibolites and migmatites [36].

Different soils develop depending on the geology and topography. Rendzic to Mollic Leptosols [37] have developed in places where carbonate rocks prevail. On acidic parent rocks, the poorly developed soils are Dystric to Umbric Regosols, which develop into Dystric to Humic

Cambisols and, at an advanced stage, into Cambic Podzols [37]. Soil formation does not take place in active scree slopes.

The SNP and its surroundings are characterized by continental inner-alpine conditions. Mean annual temperature is 5.5˚C in Scuol at 1303 m and 0.7˚C in Buffalora at 1968 m (mean 1981–2010; [38]). The corresponding mean July temperatures are 15.2˚C and 10.7˚C. Mean annual precipitation slightly increases with elevation from 706 mm in Scuol at 1303 m to 793 mm in Buffalora at 1968 m [38].

## Arthropod survey

We sampled arthropods at different elevations along the four transect lines: Val dal Botsch (hereafter B), Val Trupchun/Val Müschauns (M), Val Tavrü (T), and Val Zeznina/Macun (Z) (Fig 1, S1 Table). Two transects (B and M) are located within the SNP, while the lower part of the transect T belongs to the buffer zone of the Engiadina Val Müstair UNESCO biosphere reserve and the upper part is at the border to the SNP. The lower part of transect Z belongs to the Engiadina Val Müstair UNESCO biosphere reserve, the upper part to the SNP.

Beginning at 1800 m, we chose sampling sites along each transect line at an elevational interval of 100 m below 2000 m and at an elevational interval of 50 m above 2000 m. Transect lines mainly followed hiking paths and covered most habitat types in the SNP, ranging from alpine forests in the lower part of the valleys to alpine meadow and rocky habitats at higher elevations. To minimize disturbance of wildlife, we followed the SNP guidelines for researchers, which stipulate that sampling sites have to be within 50 m of hiking trails. The four transect lines combined comprised 65 sampling sites (S1 Table). All sampling sites are ice-free for at least 150 years.

We considered six groups of ground-dwelling arthropods: spiders (Arachnida), millipedes (Diplopoda), centipedes (Chilopoda), ants (Hymenoptera; Formicidae), ground beetles (Coleoptera; Carabidae) and rove beetles (Coleoptera; Staphylinidae). These groups are generally abundant and widely distributed, represent different trophic groups and exhibit a high phylogenetic diversity (three subphyla and four classes of arthropods).

We applied three sampling techniques to collect arthropods. Firstly, we visually searched for millipedes, centipedes, ground beetles and rove beetles on the ground, in leaf litter, and under bark, logs and stones at each sampling site for a total of 75 min. distributed over two days (hereafter active capture). Arthropods were captured with forceps and stored in ethanol (70%). Sampling was restricted to an area of approximately 30 m x 30 m or 15 m x 60 m along steep slopes. Secondly, we placed five shelters (cardboard sheets measuring 25 cm x 25 cm) fastened with some pieces of stone on the ground in each sampling site. Cardboard sheets substituted wooden boards, a frequently used method to capture millipedes [39]. Cardboard sheets were exposed for 15–30 days, after which they were carefully lifted and millipedes, centipedes, ground and rove beetles attached to or below them were captured with forceps. Thirdly, we placed five pitfall traps (plastic cups, 5.8 cm diameter) partially filled with propylene glycol in a row with an inter-trap distance of 3–5 m in each sampling site. Pitfall traps were exposed for 15–30 days, resulting in an average of 175 trap days per site. We considered all individuals of the six groups caught with the pitfall traps, cardboard sheets and visual search combined. Detailed data of the sampling effort for each site and method are presented in S1 Table.

Field work was conducted between July and September in both 2018 and 2019. We visited each sampling site twice in one of the two years (for active capture and installing and emptying the traps). Sites in the lower part of the valleys were sampled one month earlier than those in the upper part to compensate for the delayed summer at higher elevation. The methodology

for species identification is given in S1 File. The majority of the collected arthropods are deposited in the Bündner Naturmuseum in Chur (Grisons).

## Site and soil characteristics

At each sampling site, we recorded the following ecological variables: elevation (in metres above sea level), measured by a GPS receiver and checked against 1: 25 000 topographical maps (https://map.geo.admin.ch), geographical coordinates (measured with the GPS receiver), aspect (extracted from topographical maps and assigned to eight classes) and inclination (in degrees, calculated from the distance between two 20-m contour lines on topographical maps just below and above each sampling site). We assigned the vegetation type in each sampling site to one of three categories: (1) forest (trees and/or large bushes were dominant), (2) alpine grassland, and (3) patchy vegetation (mainly scree with small plant patches or single plants). Similarly, we assessed the substrate and assigned it to one of three categories: (1) developed soil (distinct horizons O and A); (2) stony debris (many stones, bedrock or inorganic soil horizons B and C visible), and (3) scree field (pieces of stone and rocks of various size).

Soil temperature can influence the composition of invertebrate communities [40]. A slope with afternoon sun is warmer than an equivalent slope with morning sun. As a proxy for soil temperature, we calculated mean heat load for each sampling site using the model of McCune and Keon [41]. This model considers topographical variables (aspect, inclination) and latitude.

At each sampling site, we collected soil samples from three randomly chosen spots at least 1 m apart from each other. Soil samples were excavated with a spade and the thickness of the organic layer and the soil mineral horizon were measured. The organic layer and the soil mineral layer (Ah horizon) were sampled separately. For analysis, we pooled the three samples of each horizon obtained in a sampling site.

Soil samples were dried at 60˚C to constant weight and sieved at 2 mm for chemical analyses. We measured total C- and N-content in milled subsamples by dry combustion using a C/N-analyzer (NC 2500, Carlo Erba Instruments, Milan, Italy). Inorganic C was removed in samples with a pH > 6.0 by fumigating with concentrated HCl (37%) prior to analysis [42]. We measured soil pH in 0.01 M $CaCl_2$ with a soil-extraction ratio of 1:2 (for mineral soil samples) and 1:4 (for organic layers) after 30 min. We determined the relative amounts of sand, clay and silt using the sedimentation method of Gee and Bauder [43]. All soil analyses were made three times using three subsamples of the pooled soil samples from each site. For data analyses, we used the mean values of the three subsamples. Data of soil characteristics and other environmental variables for each elevational band are presented in S2 Table and for each transect line in S3 Table.

## Statistical analyses

We performed all statistical analyses in the R environment (ver. 3.6.3, www.r-project.org). We considered 100-m elevational bands characterized by "band species richness" and "band relative abundance" per arthropod group as unit of data analysis. This approach allowed a comparison of data of the four transect lines, an assessment of the area of the elevational band, and reduced spatial autocorrelation among sampling sites. We used individual-based rarefied species richness to consider the large variation in number of individuals recorded among elevational bands (package iNEXT; [44]). As a proxy for the area of the different elevational bands, we measured the projected distance between the intersection of a transect line and the contour lines in steps of 100 m elevation on topographical maps (scale 1: 5000) following Sanders et al. [45]. In 100-m elevational bands on steep slopes this distance is short, in elevational bands on less steep slopes long.

In line with rarefied species richness and individual abundance, we also considered environmental variables per elevational band. We did this by averaging data from different

sampling sites for each environmental band. Environmental variables used in the analyses included elevation (midpoint of elevational band), elevational band area, aspect, inclination, heat load, vegetation type, substrate type, depth of organic layer, soil pH, C/N-ratio and clay content. Originally, we also considered the following variables: depth of mineral layer, sand and silt content, total nitrogen and total carbon content, organic and inorganic carbon content separately. However, because of intercorrelations and correlations with the C/N-ratio (in all cases $r_s > 0.40$, $P < 0.05$), we omitted these soil variables from further analyses.

We applied linear mixed models using the nlme package [46] to examine potential effects of the above-mentioned environmental variables on rarefied species richness and individual abundance separately for each arthropod group. We included transect lines as a random factor to account for the spatial autocorrelation and for the non-independence of arthropods sampled in the same transect line. For analyses, rarefied species richness and abundance were log-, sqrt- or Tuckey-transformed (type of transformation applied is given in the LME-tables in the results section). The explanatory variable elevational band area was log-transformed, inclination was sqrt-transformed, heat load, depth of organic layer and soil pH were Tukey-transformed, while C/N-ratio and clay content were arcsine sqrt-transformed. We ran the analyses twice, once with elevational band as linear term and once with elevational band as a quadratic term to detect non-linear relationships. We checked for each arthropod group which model provided the better fit using Akaike weights [47].

We used Partial Redundancy Analysis (pRDA) to determine effects of elevational band, area of elevational band, aspect, inclination, heat load, type of vegetation and substrate and soil characteristics including depth of organic layer, pH, C/N-ratio and clay content on taxonomic composition of the six ground-dwelling arthropod groups. In all cases the "species matrices" were Hellinger-transformed prior to the analysis. Model selection was conducted by forward step-wise selection from a null model containing only elevational band. In further steps, the variables of the full model including elevational band, area of elevational band, aspect, inclination, heat load, type of vegetation and substrate and soil characteristics including depth of organic layer, pH, C/N-ratio and clay content that most significantly improved the model fit were added. This process continued until no further variable improved significantly the model fit (cut-off $P = 0.05$). We checked the final model for variance inflation to detect collinearity of the variables included. Significant effects of the variables selected for the final model were tested using a post-hoc ANOVA with 999 permutations. Firstly, we used data of all species recorded for each arthropod group, and secondly, we considered a data set without singletons (species represented by only a single individual). Both analyses revealed very similar results. We therefore present only the results based on all species. All pRDA analyses were conducted in R using the vegan package [48].

To examine whether some species were indicators for one of the three vegetation types forest, alpine grassland and patchy vegetation, we calculated indicator values (IndVal; [49, 50]) using the function multipatt in the package indicspecies [51]. We set the number of permutations to 9999 and suppressed vegetation group combinations. We separately highlighted species whose IndVal was both significant and above 70%, as being good indicator species following the recommendations of [52].

## Results

### Species richness and abundance

A total of 14,782 individuals comprising 248 species were collected in the 65 sampling sites (S4 Table). Spiders were the most species-rich group (86 species), followed by rove beetles (74), ground beetles (34), millipedes (21), centipedes (19) and ants (14). Data for individual

abundance for each species per elevational band are presented in the supplementary material (S5–S10 Tables). Within sites, the total number of species recorded was positively correlated with the number of individuals sampled (spiders: $r_s$ = 0.80; millipedes: $r_s$ = 0.54; centipedes: $r_s$ = 0.56; ants: $r_s$ = 0.92; ground beetles: $r_s$ = 0.79; rove beetles: $r_s$ = 0.91; in all groups: n = 65, P > 0.0001). The number of singleton species (only a single individual sampled) ranged from 0 in millipedes, 1 in ants (7.1% of the species recorded), 3 in centipedes (15.7%), 5 in ground beetles (14.7%), 15 in rove beetles (20.2%) to 22 in spiders (25.6%).

## Comparison of linear vs. quadratic (mid-domain) elevational effects

To examine whether elevation-related changes in rarefied species richness and abundance were linear or quadratic (corresponding to MDE influence), we compared the fits of different models with elevation as either linear or quadratic variable using the AIC-weight approach [47]. Considering rarefied species richness, LME models with a quadratic term for elevation had a better fit than models with a linear term in most groups (spiders, millipedes, ants, rove beetles), indicating a mid-domain effect (Table 1). Centipedes were an exception, for which the LME model with the linear term for elevation provided the better fit (Table 1). For ground beetles, we found no influence of elevation on rarefied species richness in either model (Table 1).

The findings for abundance mirrored those for rarefied species richness in most groups. LME models with a quadratic term for elevation provided the better fit, indicating a mid-domain effect for spiders, ants and rove beetles, while for centipedes the model with the linear term had the better fit (Table 1). In millipedes and ground beetles, there was no influence of elevation on abundance in either model (Table 1).

## Rarefied species richness

Considering LME-models with a quadratic fit of elevational band, elevation influenced rarefied species richness in spiders, millipedes, centipedes, ants and rove beetles, but not in ground beetles (Table 2, Fig 2), although the fit with the linear elevation term was better in centipedes (Table 1). The relationship between rarefied species richness and elevation was hump-shaped in spiders and millipedes, suggesting a mid-peak at low elevation (maximum at 2150 m in spiders and 2250 m in millipedes; Fig 2). Similarly, the patterns observed in ants and rove beetles indicate a low elevation plateau (1850 m to 2150 m in ants, 1850 m to 2250 m in rove beetles), followed by a decrease in rarefied species richness at higher elevation (Fig 2). In centipedes, a linear decrease in rarefied species richness with elevation was observed, while in ground beetles no effect of elevation on rarefied species richness was found (Fig 2). The area of the elevational bands, however, did not influence this finding (Table 2).

Aspect affected rarefied species richness of centipedes, ants and rove beetles (Table 2). Rarefied species richness of centipedes and ants was increased in sites facing to east and south-east (S1 Fig). Rarefied species richness of rove beetles showed peaks in sites exposed to north-east and north-west (S1 Fig). Soil pH influenced rarefied species richness of millipedes, ground beetles and rove beetles (Table 2). In rove beetles, rarefied species richness decreased with increasing soil pH (S2 Fig), while in millipedes and ground beetles this relationship was hump-shaped (S2 Fig). Rarefied species richness of ants was affected by soil clay content (Table 2), with number of species decreasing with increasing clay content. The other environmental variables did not affect rarefied species richness of the six groups examined (Table 2). In centipedes, the LME-model with the linear elevation term revealed similar findings for the environmental variables than the model with the quadratic term (S11 Table).

**Table 1. Model fit (AIC) comparing linear elevation effects with quadratic elevation effects (MDE) on rarefied species richness and abundance of six groups of arthropods.**

| | AIC | ΔAIC | Weight |
|---|---|---|---|
| **Rarefied species richness** | | | |
| **Spiders** | | | |
| Linear model | 137.5 | 0.69 | 0.415 |
| MDE model | **136.8** | **0.00** | **0.585** |
| **Millipedes** | | | |
| Linear model | 261.1 | 1.92 | 0.277 |
| MDE model | **259.2** | **0.00** | **0.723** |
| **Centipedes** | | | |
| Linear model | **61.8** | **0.00** | **0.521** |
| MDE model | 62.0 | 0.17 | 0.479 |
| **Ants** | | | |
| Linear model | 119.0 | 1.95 | 0.273 |
| MDE model | **117.1** | **0.00** | **0.727** |
| **Ground beetles** | | | |
| Linear model | No elevation effect | | |
| MDE model | No elevation effect | | |
| **Rove beetles** | | | |
| Linear model | 149.6 | 2.73 | 0.203 |
| MDE model | **146.8** | **0.00** | **0.797** |
| **Individual abundance** | | | |
| **Spiders** | | | |
| Linear model | 129.5 | 0.79 | 0.402 |
| MDE model | **128.7** | **0.00** | **0.598** |
| **Millipedes** | | | |
| Linear model | No elevation effect | | |
| MDE model | No elevation effect | | |
| **Centipedes** | | | |
| Linear model | **209.2** | **0.00** | **0.589** |
| MDE model | 209.9 | 0.72 | 0.411 |
| **Ants** | | | |
| Linear model | 221.4 | 8.39 | 0.015 |
| MDE model | **213.0** | **0.00** | **0.985** |
| **Ground beetles** | | | |
| Linear model | No elevation effect | | |
| MDE model | No elevation effect | | |
| **Rove beetles** | | | |
| Linear model | 161.4 | 1.85 | 0.284 |
| MDE model | **159.2** | **0.00** | **0.716** |

## Abundance

Considering LME-models with a quadratic fit of elevational band, individual abundance was influenced by elevation in spiders, ants and rove beetles, but not in the other groups (Table 3, Fig 3). The relationship between individual abundance (measured by the number of individuals captured in each group) and elevation was hump-shaped in spiders and ants (Fig 3). In rove beetles, this relationship showed a low-elevation plateau (1850 to 2150 m), followed by a decrease at higher elevations (Fig 3). Surprisingly, in centipedes, individual abundance

**Table 2. Summary of the linear mixed models (LME) examining the effects of elevational band (quadratic fit, corresponding to a mid-domain effect), area of elevational band, aspect, inclination, heat load, type of vegetation and substrate and soil characteristics including depth of organic layer (cm), pH, C/N-ratio and clay content (%) on rarefied species richness of six arthropod groups.**

| | Rarefied species richness | | | | | |
| --- | --- | --- | --- | --- | --- | --- |
| | Spiders [1] | Millipedes [2] | Centipedes [3] | Ants [1] | Ground beetles [2] | Rove beetles [3] |
| Elevational band | $F_{1,23} = 6.33$, **P = 0.019** | $F_{1,32} = 21.22$, **P < 0.0001** | $F_{1,24} = 8.20$, **P = 0.009** | $F_{1,20} = 34.22$, **P < 0.0001** | $F_{1,22} = 1.19$, P = 0.29 | $F_{1,23} = 42.07$, **P < 0.0001** |
| Elevational band area [1] | – | – | $F_{1,24} = 1.57$, P = 0.22 | – | $F_{1,22} = 1.19$, P = 0.29 | $F_{1,23} = 1.75$, P = 0.20 |
| Aspect | $F_{6,23} = 1.87$, P = 0.13 | – | $F_{6,24} = 2.62$, **P = 0.044** | $F_{6,20} = 4.73$, **P = 0.004** | $F_{6,22} = 1.42$, P = 0.25 | $F_{6,23} = 4.34$, **P = 0.005** |
| Inclination [2] | $F_{1,23} = 3.11$, P = 0.09 | – | – | $F_{1,20} = 1.79$, P = 0.20 | – | $F_{1,23} = 3.13$, P = 0.09 |
| Heat load [3] | $F_{1,23} = 2.96$, P = 0.10 | – | – | $F_{1,20} = 2.58$, P = 0.12 | $F_{1,22} = 3.04$, P = 0.09 | – |
| Vegetation type | $F_{1,23} = 1.60$, P = 0.22 | – | – | $F_{2,20} = 2.05$, P = 0.15 | – | – |
| Substrate type | – | – | – | – | $F_{2,22} = 1.97$, P = 0.16 | – |
| Organic layer [3] | – | – | – | $F_{1,20} = 1.17$, P = 0.29 | – | – |
| Soil pH [3] | – | $F_{1,32} = 5.51$, **P = 0.025** | $F_{1,24} = 2.13$, P = 0.16 | – | $F_{1,22} = 4.52$, **P = 0.045** | $F_{1,23} = 17.36$, **P < 0.001** |
| Soil C/N-ratio [4] | – | – | – | $F_{1,20} = 1.92$, P = 0.18 | – | – |
| Soil clay content [4] | – | – | $F_{1,24} = 3.12$, P = 0.09 | $F_{1,20} = 4.15$, **P = 0.048** | – | $F_{1,23} = 3.97$, P = 0.06 |

Significant P-values (P < 0.05) are in bold

"–"variable was excluded from the model due to the step-wise model reduction procedure

1 = log-transformed

2 = sqrt-transformed

3 = Tukey-transformed

4 = arcsine-sqrt-transformed

increased linearly with increasing elevation (Fig 3). In millipedes and ground beetles, no effect of elevation on individual abundance was found (Table 3, Fig 3).

Similar to rarefied species richness, the area of elevational bands did not influence the abundance of any of the arthropod groups examined. However, aspect affected the abundance of millipedes, ants and rove beetles (Table 3). Ant abundance was highest in sites exposed to east, south-east and south-west (S3 Fig). Millipede abundance was highest in sites facing to south-east, while that of rove beetles was highest in sites exposed to east and north-east (S3 Fig). Soil pH influenced the abundance of spiders, centipedes and rove beetles (Table 3). Similar to rarefied species richness, abundance of rove beetles decreased with increasing soil pH (S4 Fig). In spiders and centipedes, this relationship was U-shaped (S4 Fig). Vegetation type affected the abundance of spiders and that of ground beetles (Table 3), being lowest in sites with patchy vegetation (S5 Fig). Furthermore, heat load influenced the individual abundance of spiders and soil clay content that of ants (Table 3).

In centipedes, the LME-model with the linear elevation term (S12 Table) and the model with the quadratic term revealed similar findings for the environmental variables.

## Species composition

The four valleys harboured similar species assemblages in all arthropod groups examined, indicated by Sørensen's similarity coefficients (B: mean of all groups = 0.554; M: 0.585; T: 0.608; Z: 0.598). The overall similarity in species assemblages among the four valleys was highest in centipedes (mean = 0.803), followed in decreasing order by millipedes (0.704), ground beetles (0.631), ants (0.576), rove beetles (0.460) and spiders (0.371).

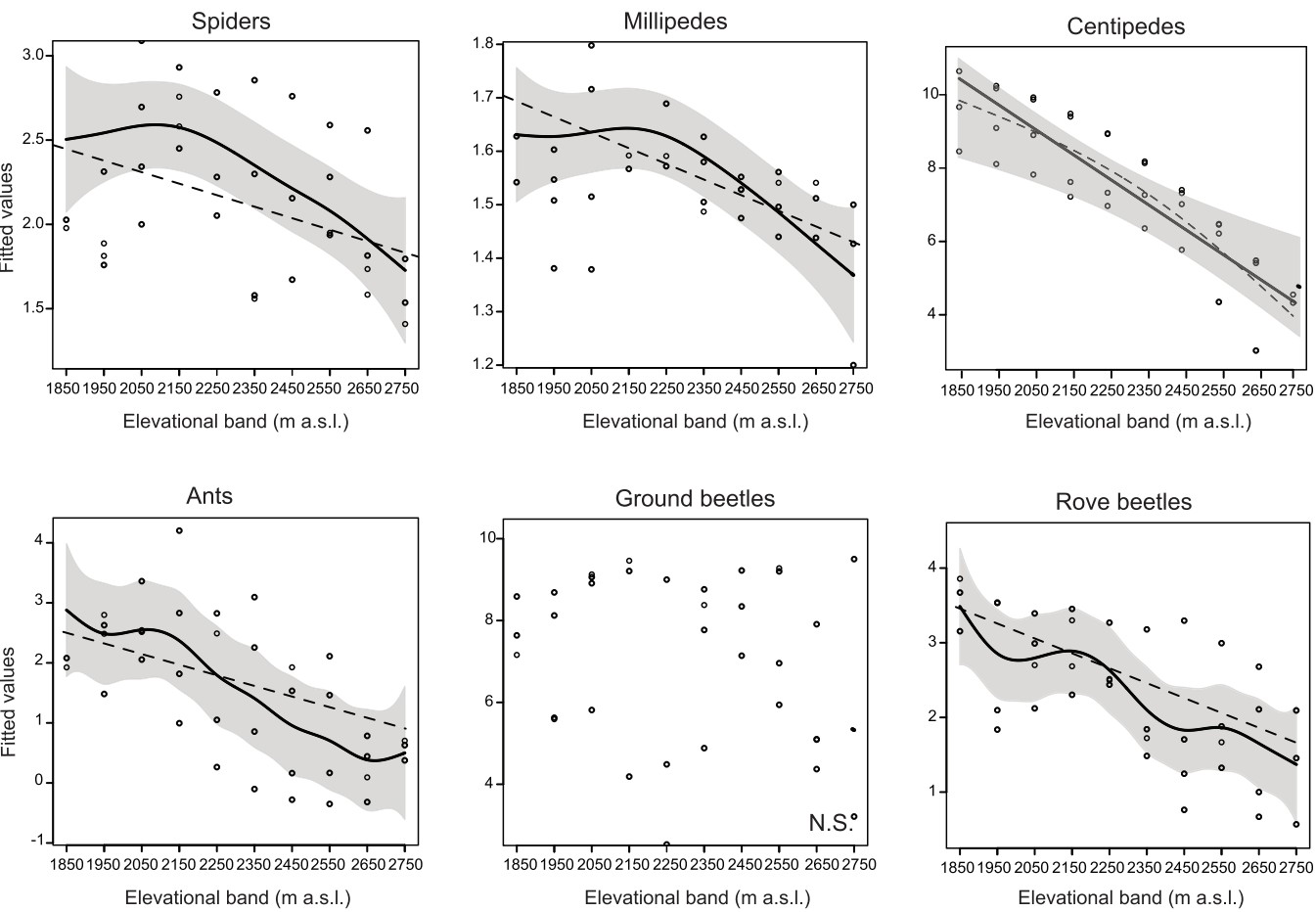

**Fig 2. Relationships between rarefied species richness at the elevational band level of six groups of ground-dwelling arthropods and elevation.** Data for the four replicate transect lines (mountain slopes) are shown as circles. Fitted values of linear and MDE-models are displayed (the model with the better fit is depicted with a bold continuous line and a 95% confidence interval shown in grey). ns indicates that both models were not significant.

Partial Redundancy Analysis (pRDA) revealed that species composition of the different groups was more frequently influenced by topographic characteristics than by soil characteristics. Species composition was affected by elevation in all arthropod groups examined (Table 4). In all groups, the change was expressed along the first axis (Fig 4). The area of the elevational bands, however, influenced species composition only in spiders and centipedes (Table 4). Heat load affected species composition in all arthropod groups, while aspect per se had only an effect on species composition in centipedes, ants and rove beetles, and inclination in millipedes (Table 4). Among the soil characteristics, soil pH influenced species composition in centipedes, ants and rove beetles (Table 4). Species composition of ants was also affected by the depth of the organic layer and that of rove beetles by the soil C/N-ratio (Table 4).

Indicator value analysis revealed five spider, six millipede, two centipede, three ant, five ground beetle and eight rove beetle species as indicators for forest (S13 Table). However, only two millipede, one ant, two ground beetle and two rove beetle species reached an IndVal of 70% or more (S13 Table). The only indicator species for grassland was a ground beetle species with an IndVal of 59%. Considering patchy vegetation, two spider, one millipede and two centipede species had a significant IndVal, though only for one millipede and one centipede species the IndVal was > 70% (S13 Table).

**Table 3. Summary of the linear mixed models (LME) examining the effects of elevational band (quadratic fit, corresponding to a mid-domain effect), area of elevational band, aspect, inclination, heat load, type of vegetation and substrate and soil characteristics including depth of organic layer (cm), pH, C/N-ratio and clay content (%) on the number of individuals belonging to six arthropod groups.**

| | Number of individuals | | | | | |
|---|---|---|---|---|---|---|
| | Spiders [3] | Millipedes [1] | Centipedes [3] | Ants [3] | Ground beetles [1] | Rove beetles [1] |
| Elevational band | $F_{1,23} = 12.34$, **P = 0.002** | $F_{1,25} = 0.01$, P = 0.98 | $F_{1,19} = 3.26$, P = 0.09 | $F_{1,20} = 34.22$, **P <0.0001** | $F_{1,29} = 2.73$, P = 0.11 | $F_{1,21} = 40.05$, **P <0.0001** |
| Elevational band area [1] | – | $F_{1,25} = 1.13$, P = 0.30 | – | – | – | – |
| Aspect | $F_{6,23} = 1.89$, P = 0.13 | $F_{6,25} = 3.56$, **P = 0.011** | $F_{6,19} = 1.99$, P = 0.13 | $F_{6,20} = 4.73$, **P = 0.004** | – | $F_{6,21} = 5.09$, **P = 0.002** |
| Inclination [2] | – | – | – | $F_{1,20} = 1.79$, P = 0.20 | | $F_{1,21} = 3.50$, P = 0.08 |
| Heat load [3] | $F_{1,23} = 9.21$, P = **0.006** | $F_{1,25} = 1.22$, P = 0.28 | $F_{1,19} = 1.30$, P = 0.27 | $F_{1,20} = 2.58$, P = 0.12 | – | – |
| Vegetation type | $F_{1,23} = 5.75$, P = **0.009** | – | $F_{2,19} = 1.44$, P = 0.26 | $F_{1,20} = 2.05$, P = 0.15 | $F_{2,29} = 3.43$, **P = 0.046** | $F_{2,21} = 2.51$, P = 0.10 |
| Substrate type | – | – | $F_{2,19} = 1.65$, P = 0.22 | – | $F_{2,29} = 2.49$, P = 0.11 | $F_{2,21} = 2.17$, P = 0.14 |
| Organic layer [3] | – | – | – | – | – | – |
| Soil pH [3] | $F_{1,23} = 4.92$, P = **0.037** | – | $F_{1,19} = 8.78$, **P = 0.008** | – | – | $F_{1,21} = 14.17$, **P = 0.001** |
| Soil C/N-ratio [4] | – | – | $F_{1,19} = 0.91$, P = 0.35 | $F_{1,20} = 1.92$, P = 0.18 | – | – |
| Soil clay content [4] | – | – | $F_{1,19} = 3.82$, P = 0.07 | $F_{1,20} = 4.42$, **P = 0.049** | – | – |

Significant P-values (P < 0.05) are in bold

"–"variable was excluded from the model due to the step-wise model reduction procedure

1 = log-transformed

2 = sqrt-transformed

3 = Tukey-transformed

4 = arcsine-sqrt-transformed

## Discussion

Our study showed that rarefied species richness in five out of the six ground-dwelling arthropod groups changed with elevation, although the responses corresponded to three different patterns (linear decrease, low elevation plateau followed by a decrease, and midpoint peak). These patterns were only partially mirrored when considering individual abundance. Generally, topographical environmental factors were more important in explaining patterns of species richness and individual abundance in the six arthropod groups than local soil and habitat characteristics. Elevation influenced species composition in all groups examined.

### Patterns of species richness

A decrease of species richness along elevational gradients has been previously reported for numerous taxa from across the globe [9, 22]. In our study, we confirmed this decrease for five groups of ground-dwelling arthropods with mostly low mobility. The similarities in habitat use in the ectotherm groups studied by us may suggest similar effects of the changing environmental variables along the elevational gradients on species richness. Interestingly, however, we found three different patterns of species richness change on the same mountain slopes. In only one group (ground beetles), we recorded no significant pattern of species richness with elevation. The three patterns observed (low elevation plateau followed by a decrease, mid-peak, and linear decrease) have in common that they show a decrease in species richness at higher elevations. Our transect lines started at 1800 m a.s.l., slightly above the elevation of the main valley (river Inn). Thus, our data set is truncated at the lower end of the transects. It is therefore

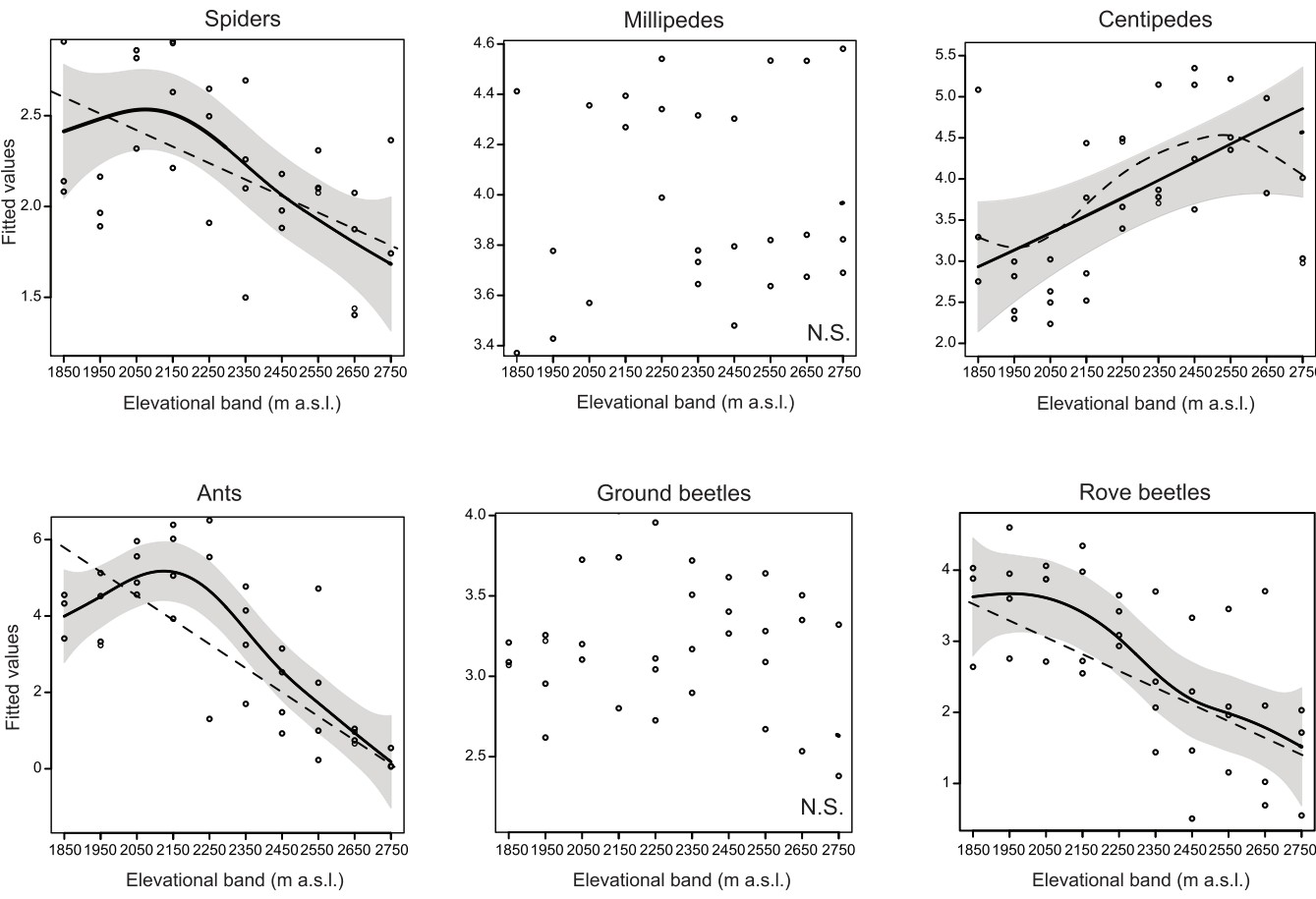

**Fig 3. Relationships between individual abundance at the elevational band level of six groups of ground-dwelling arthropods and elevation.** Data for the four replicate transect lines (mountain slopes) are shown as circles. Fitted values of linear and MDE-models are displayed (the model with the better fit is depicted with a bold continuous line and a 95% confidence interval shown in grey). ns indicates that both models were not significant.

possible that the observed patterns of low-elevation plateaus (in ants and rove beetles) and linear decrease (in centipedes) are part of mid-peak patterns with a relatively low peak.

The mid-peak pattern in species richness has frequently been explained by geometric constraints (mid-domain effect; MDE [29, 53]). MDE stresses that the random distribution of ranges within a restricted geographical domain (e.g. the lower and upper elevational limit on a mountain slope) without the need for any climatic or environmental factors for the placement of the species ranges leads to mid-peaks in species [29, 53]. MDE has been demonstrated in numerous empirical studies for several taxa (e.g. in plants: [53], moths: [28], gastropods: [54]). It should be noted that in our study the lower boundary of some species' ranges was most probably not assessed. This could move the mid-peak slightly downslope.

Alternatively, a mid-peak pattern may result if intense human land use at low elevation reduces species richness [9]. The diversity of arthropods at low elevations is often heavily impacted by intense land-use practices (e.g. [55]), which may influence the findings on biodiversity along elevational transects in studies starting at low elevations (e.g. [56]). However, this explanation is not valid for our study because the majority of sampling sites were in a strictly protected area (since 1914), while the remaining sites were in extensively managed cattle pastures. Mid-peak patterns can also result from elevation-dependent patterns in precipitation and primary productivity or from biotic interactions among species [9]. For example, species

**Table 4. Summary of Partial Redundancy Analysis (partial RDA) examining the effects of elevational band, area of elevational band, aspect, inclination, heat load, type of vegetation, and substrate and soil characteristics including depth of organic layer (cm), pH, C/N-ratio and clay content (%) on species composition of six arthropod groups.**

| | Species composition | | | | | |
| --- | --- | --- | --- | --- | --- | --- |
| | **Spiders** | **Millipedes** | **Centipedes** | **Ants** | **Ground beetles** | **Rove beetles** |
| Elevational band | $F_{1,34} = 4.15$, **P = 0.001** | $F_{1,34} = 8.16$, **P = 0.001** | $F_{1,32} = 3.55$, **P = 0.003** | $F_{1,32} = 5.38$, **P = 0.001** | $F_{1,35} = 8.55$, **P = 0.001** | $F_{1,32} = 3.34$, **P = 0.001** |
| Elevational band area [1] | $F_{1,34} = 2.99$, **P = 0.001** | – | $F_{1,32} = 2.10$, **P = 0.031** | – | – | – |
| Aspect | – | – | $F_{1,32} = 3.03$, **P = 0.006** | $F_{1,32} = 3.63$, **P = 0.007** | – | $F_{1,32} = 2.59$, **P = 0.001** |
| Inclination [2] | – | $F_{1,34} = 2.21$, **P = 0.032** | – | – | – | – |
| Heat load [3] | $F_{1,34} = 2.57$, P = **0.003** | $F_{1,34} = 4.03$, **P = 0.001** | $F_{1,32} = 2.25$, **P = 0.026** | $F_{1,32} = 3.48$, **P = 0.015** | $F_{1,35} = 5.67$, **P = 0.001** | $F_{1,32} = 2.15$, **P = 0.010** |
| Vegetation type | – | – | – | – | – | – |
| Substrate type | – | – | – | – | – | – |
| Organic layer [3] | – | – | – | $F_{1,32} = 4.78$, **P = 0.003** | – | – |
| Soil pH [3] | – | – | $F_{1,32} = 4.88$, **P = 0.001** | $F_{1,32} = 2.95$, **P = 0.022** | – | $F_{1,32} = 3.87$, **P = 0.001** |
| Soil C/N-ratio [4] | – | – | – | – | – | $F_{1,32} = 2.13$, **P = 0.010** |
| Soil clay content [3] | – | – | – | – | – | – |

Significant P-values (P < 0.05) are in bold

"–"variable was excluded from the RDA due to the reduced model procedure

1 = log-transformed

2 = sqrt-transformed

3 = Tukey-transformed

4 = arcsine-sqrt-transformed

richness patterns in ants are strongly influenced by competitive interactions among species and dominance hierarchies [57, 58].

Similar to the low-elevation plateau pattern observed in ants and rove beetles, the pattern of linear decrease found for centipedes may represent the upper part of a mid-peak pattern. However, it is possible that species richness of centipedes linearly decreases with elevation, which could be caused by changes in environmental factors (e.g. temperature) with elevation. Indeed, centipede species distributions are to a certain degree affected by temperature [59]. A pattern of linear decrease may also result if most distributions of species are restricted to lower elevations as has been shown for the distributions of centipedes in the nearby Eastern Italian Alps [60]. Furthermore, a pattern of linear decrease in species richness could be a result of a shorter period available for recolonisation after the retreat of glaciers at higher elevation [61]. However, this does not apply for the transect lines examined in our study, because all sampling sites are ice-free for at least 150 years.

A number of factors have been implicated as underlying causes of elevational species diversity gradients [9]. In our study, elevation *per se* had the largest impact on rarefied species richness (as well as on individual abundance and species composition) in the six arthropod groups examined (in ground beetles only on species composition). Elevation acts as a proxy for several intercorrelated variables (temperature, air pressure, solar radiation, length of vegetation period) and is further connected to site-specific environmental factors (precipitation, soil characteristics, habitat type; [9, 12, 13]). These factors are difficult to disentangle on a given mountain slope. Thus, the consistent impact of elevation could be the result of the various factors listed above affecting the arthropod groups examined in different ways.

Interestingly, in our study, models with a quadratic term for elevation had in most cases a better fit than models with a linear term for elevation. A quadratic term for elevation is more suited to describe two common elevational patterns of diversity, namely low elevation plateau

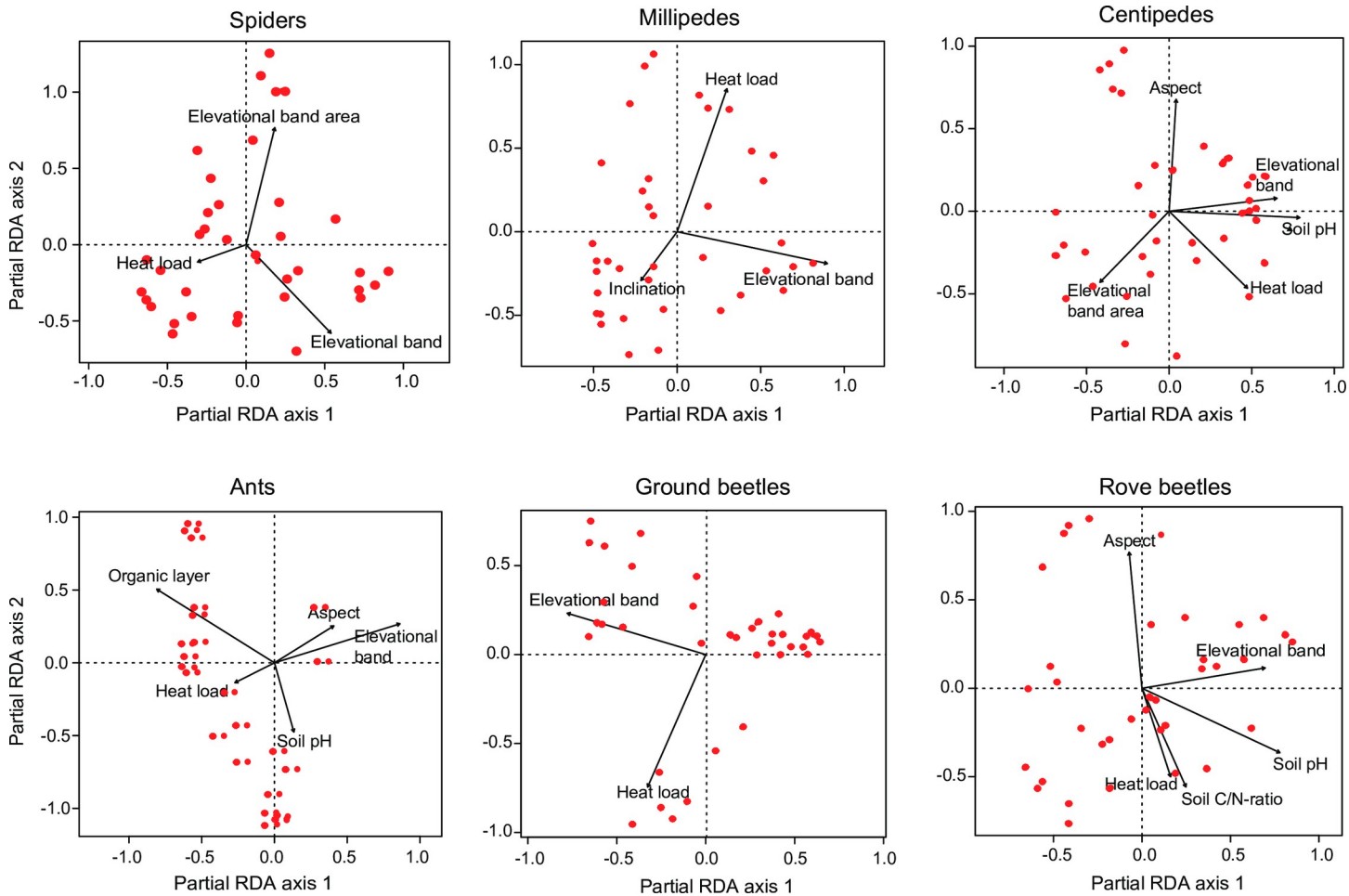

**Fig 4. Results of Partial Redundancy Analysis (pRDA) showing the relationship of species composition of six groups of ground-dwelling arthropods along four elevational transects in the Swiss National Park to topographical, soil and other environmental characteristics.** Red dots represent the elevational bands of the four transect lines. Corresponding statistics are shown in Table 4.

followed by a decrease and mid-peak patterns [9, 58], which were also frequently found in our study.

Available land area is considered a major driver of species richness on mountain slopes [22–25]. Due to the conical shape of most mountains, land area is shrinking with increasing elevation, thereby narrowing opportunities for life [22, 26]. In our study, we did not find any effect of elevational band area on rarefied species richness or individual abundance in any of the arthropod groups examined. However, species composition in spiders and centipedes was influenced by elevational band area. There are few studies examining the effect of elevational band area on arthropod diversity. An example is Sanders [23], who reported that ant species richness increased with elevational band area in each of three states in the western United States examined, contrasting our findings.

We expected that local environmental factors (soil characteristics, substrate structure, and vegetation type) play an important role for the diversity of the groups examined. Most species of the six arthropod groups are characterized by relatively low mobility and a ground-dwelling life style. This is also true for groups containing non-ground dwelling species as pitfall traps were geared towards ground surface-active species. Interestingly, however, soil characteristics,

with the exception of pH, seem to have little effect on rarefied species richness in most of the groups examined. A possible explanation for the lack of explanatory power of soil characteristics is that they are highly intercorrelated with elevation [62].

Species richness of millipedes, ground beetles and rove beetles were influenced by soil pH. In millipedes the relationship was hump-shaped. This is surprising as these animals have a calcified exoskeleton and therefore are supposed to benefit from soils containing carbonates [63]. Furthermore, more millipede species are known to occur on calcareous than on acidic soils [64], though some species exhibit higher activity on acidic soils [15]. Like for millipedes, the relationship between ground beetle species richness and soil pH was hump-shaped. Ground beetle species are known to differ regarding their soil pH preferences [65], which in turn shapes ground beetle communities [66–68]. However, different patterns have been reported (increasing species richness with pH: [66], decreasing with pH: [67]). In our study, soil pH negatively affected species richness of rove beetles. Soil pH is assumed to indirectly influence rove beetle communities by changing the quality of their prey [69].

## Patterns of individual abundance

In general, individual abundance patterns in arthropods have received little attention along elevational gradients in temperate mountain regions. Possible explanations include difficulties for standardized quantitative sampling at high elevations, different peak activities at different elevations (seasonality) and assumed low abundance (and species richness) in such extreme environments. In mountain areas that experience frost in the winter, it is assumed that many arthropods spend this period in a state of diapause [70], resulting in a short synchronous activity peak in summer [71]. We circumvented the problem of changing peak activity with elevation by sampling at higher elevations one month later than at lower elevations. In tropical mountains (e.g. [72]), arthropods are exposed to different seasonal fluctuations in temperature and precipitation, which complicates comparisons with patterns found in our study.

Elevational patterns for abundance were similar to those for rarefied species richness for spiders (mid-peak) and rove beetles (low elevation plateau followed by decrease), but not for millipedes, centipedes and ants. The mid-peak pattern in ant abundance could be a result of dominance hierarchies [57]. In our study, some dominant species (e.g. *Formica lugubris* Zetterstedt, 1838 and *Formica sanguinea* Latreille, 1798; [73]) did not occur at high elevations. Therefore, some sub-dominant species may have become more abundant (more or larger colonies) in sites with none of the dominant species present, before also declining at very high elevations.

Our findings of different patterns for different taxonomical groups in the same region are in line with the results of Winkler et al. [74], who showed that the abundance of beetles and spiders was affected by elevation in the Central Alps, while this was not the case in springtails and oribatid mites. In our study, millipede abundance showed no significant pattern with elevation. This can be partly explained by the great abundance of a high-alpine specialist millipede species (*Pterygophorosoma alticolum* (Verhoeff, 1894)), which may mask an elevational abundance decrease in other species. For ground beetles, we found no elevational pattern in abundance, confirming Jung et al. [75] and Pizzolotto et al. [76], but contrasting Röder et al. [72], who observed a hump-shaped pattern in the tropics and Zou et al. [77], who reported an increase in ground beetle species' abundance with increasing elevation on mountain slopes in north-east China. In our study, centipede abundance likewise increased with elevation, even though rarefied species richness decreased with elevation.

It should also be noted that within a taxonomical group, subgroups may respond differently to elevational gradients. Contrasting patterns of subgroups may be a reason for the absence of abundance and species richness patterns for ground beetles in our study. Subgroups could be

taxonomical or be defined by traits. For example, Şenyüz et al. [78] reported different patterns for species richness and abundance for small vs. large Scarabaeinae (dung beetles) along an elevational gradient in Turkey.

Habitat-specific characteristics such as slope stability and vegetation cover may play an important role in determining centipede abundance in this region of the Alps [60]. Likewise, vegetation structure and plant diversity are known to influence ground beetle and spider abundance [79–82]. Most alpine ant species construct nests in the soil or close to the surface and many species show specific preferences for soil types [73]. This may explain our finding that ant abundance and species richness were affected by clay content in the soil.

## Species composition

In our study, we recorded changes in species composition with elevation in all groups examined (Fig 4). This matches findings by other studies on arthropods along elevation gradients [5, 77, 83, 84]. Such changes in species composition along mountain slopes could be attributed to various environmental factors, including climatic variables such as temperature and precipitation but also to the structure and type of the vegetation, and to soil properties [85–88]. In our study, in addition to elevation (as a proxy for several environmental factors), heat load also influenced the species composition in all groups. Heat load combines aspect and inclination (and latitude, which is of minor importance at the scale of our study) and is not correlated with elevation. Thus, heat load represents components of thermal energy available at study sites not explained by elevation. Aspect per se had an effect on species composition in centipedes, ants and rove beetles, while inclination affected species composition in millipedes. Furthermore, soil pH affected species composition of centipedes, ants and rove beetles in our study. As explained above, the distributions of some species are related to soil pH, which also shapes the composition of local assemblages.

Considering multi-taxa approaches along the same slope, varied patterns in species diversity among different taxa have been previously reported (e.g. [5, 10, 30, 32, 56]). Furthermore, within the same taxonomical group, different patterns have been demonstrated on different mountain slopes (e.g. in ants [58] and in ground beetles [89]). In our study, we aimed to analyse general patterns for six different arthropod groups by combining data from four mountain slopes (transect lines).

## Baseline data for studies of environmental change

Mountain species are affected by ongoing climate warming, which is altering the species' distributions, resulting frequently in uphill shifts [90–93]. This can lead to a reduction of available habitat for high-elevation species. Furthermore, the specialist mountain species will be affected by advancing species from lower elevations, which alter biotic interactions in high mountain ecosystems, including competition [94–96]. The temperature increase in the European Alps is above average [97, 98]. Repeated transect studies considering multiple taxa conducted in regularly spaced sites and with consistent sampling protocols offer a valuable tool to analyse the impact of climate warming on ground-dwelling invertebrate communities. Our survey thus provides valuable baseline data.

## Conclusions

Our study, focusing on six groups of ground-dwelling arthropods with many species of relatively low mobility, confirmed the variability in patterns of elevational species richness even for taxonomical groups with similar lifestyles [5, 10], and highlights the need for multi-taxa approaches. Interestingly, despite the variability found in our study among taxonomical

groups, the general finding was an overall strong effect of elevation per se on arthropod diversity, while most local soil and vegetation characteristics appeared to play a minor role.

The SNP is a strongly protected area for more than 100 years. Thus, the species assemblages recorded at the various sites are not affected by direct human interaction. This contrasts with other studies, where historical and management-related factors may have more heavily influenced current species distributions.

## Supporting information

**S1 File. Methodology, nomenclature and literature used for species identification.**
(DOCX)

**S1 Fig. Effect of slope aspect on rarefied species richness of centipedes, ants and rove beetles.**
(PDF)

**S2 Fig. Relationships between soil pH and rarefied species richness of millipedes, ground beetles and rove beetles.**
(PDF)

**S3 Fig. Effect of slope aspect on individual abundance of millipedes, ants and rove beetles.**
(PDF)

**S4 Fig. Relationships between soil pH and individual abundance of spiders, centipedes and rove beetles.**
(PDF)

**S5 Fig. Effect of vegetation type on individual abundance of spiders and ground beetles.**
(PDF)

**S1 Table. Sampling effort for each method and site.**
(XLS)

**S2 Table. Species richness, individual abundance and rarefied species richness for six groups of arthropods and environmental variables at elevational band level.**
(XLSX)

**S3 Table. Soil characteristics in four transects in the SNP and its surroundings at site level.**
B – Val dal Botsch; M–Val Trupchun/Val Müschauns; T–Val Tavrü, and Z–Val Zeznina/ Macun. Mean values and ranges are presented for each transect line; n indicates the number of sampling sites.
(DOC)

**S4 Table. Individual abundance and species (in parenthesis) in different ground-dwelling arthropod groups recorded in each of four transect lines in the SNP and its surroundings.**
(DOC)

**S5 Table. Individual abundance for each spider species at the elevational band level.**
(XLSX)

**S6 Table. Individual abundance for each millipede species at the elevational band level.**
(XLSX)

**S7 Table. Individual abundance for each centipede species at the elevational band level.**
(XLSX)

**S8 Table. Individual abundance for each ant species at the elevational band level.**
(XLSX)

**S9 Table. Individual abundance for each ground beetle species at the elevational band level.**
(XLSX)

**S10 Table. Individual abundance for each rove beetle species at the elevational band level.**
(XLSX)

**S11 Table. Summary of the linear mixed models (LME) examining the effects of elevation (elevational bands), area of elevational bands, and local environmental characteristics on the number of rarefied species richness of six arthropod groups.**
(DOCX)

**S12 Table. Summary of the linear mixed models (LME) examining the effects of elevation (elevational bands), area of elevational bands, and local environmental characteristics on the number of individuals belonging to six arthropod groups.**
(DOCX)

**S13 Table. Number of indicator species in each arthropod group for three vegetation types.**
(DOCX)

## Acknowledgments

The SNP direction and SNP Research Commission granted permission for fieldwork in the SNP and provided logistical help. The authors thank C. Rossi, S. Wiesmann, S. Campell Andri and S. Bunte (all staff members of SNP) for advice and for helping during the fieldwork, and T. Estermann (SNP) for producing the map. The authors are thankful to A. Criado, C. Bonetti, E. Cuesta, H. Eggenberger, I. Bobbitt, I. Gilgado, J. Meltzer, L. Yapura, M. Baur, R. Lázaro, R. Galán, S. Friry, S. Meyer, M. Zimmermann, J. Fetzer and V. Martínez-Pillado for assistance during fieldwork and in the laboratory, and É. Cudré-Mauroux, E. Pandiamakkal and L. Yapura for sorting the arthropods sampled in the traps. We are grateful to A. Szallies (rove beetles), D. Cabanillas (centipedes) and S. Buchholz (spiders) for species determination. We thank P. A. V. Borges and an anonymous reviewer for valuable comments on the manuscript.

## Author Contributions

**Conceptualization:** José D. Gilgado, Bruno Baur.

**Data curation:** José D. Gilgado, Hans-Peter Rusterholz.

**Formal analysis:** Hans-Peter Rusterholz.

**Funding acquisition:** José D. Gilgado, Bruno Baur.

**Investigation:** José D. Gilgado, Brigitte Braschler, Stephan Zimmermann, Yannick Chittaro, Bruno Baur.

**Methodology:** José D. Gilgado, Stephan Zimmermann, Bruno Baur.

**Project administration:** José D. Gilgado, Bruno Baur.

**Writing – original draft:** José D. Gilgado, Hans-Peter Rusterholz, Brigitte Braschler, Bruno Baur.

**Writing – review & editing:** José D. Gilgado, Hans-Peter Rusterholz, Brigitte Braschler, Stephan Zimmermann, Yannick Chittaro, Bruno Baur.

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
