## [Decision Letter · Decision Letter 0]

6 Jun 2022

PONE-D-22-06634Six groups of ground-dwelling arthropods show different diversity responses along elevational gradients in the Swiss AlpsPLOS ONE

Dear Dr. Braschler,

Thank you for submitting your manuscript to PLOS ONE. After careful consideration, we feel that it has merit but does not fully meet PLOS ONE’s publication criteria as it currently stands. Therefore, we invite you to submit a revised version of the manuscript that addresses the points raised during the review process.

We look forward to receiving your revised manuscript.

Kind regards,

Bi-Song Yue, Ph.D

Academic Editor

PLOS ONE

Journal Requirements:

2. We note that Figure 1 in your submission contain [map/satellite] images which may be copyrighted. All PLOS content is published under the Creative Commons Attribution License (CC BY 4.0), which means that the manuscript, images, and Supporting Information files will be freely available online, and any third party is permitted to access, download, copy, distribute, and use these materials in any way, even commercially, with proper attribution. For these reasons, we cannot publish previously copyrighted maps or satellite images created using proprietary data, such as Google software (Google Maps, Street View, and Earth). For more information, see our copyright guidelines: http://journals.plos.org/plosone/s/licenses-and-copyright.

Reviewers' comments:

Reviewer's Responses to Questions

**Comments to the Author**

1. Is the manuscript technically sound, and do the data support the conclusions?

Reviewer #1: Yes

Reviewer #2: Yes

2. Has the statistical analysis been performed appropriately and rigorously? 

Reviewer #1: Yes

Reviewer #2: Yes

3. Have the authors made all data underlying the findings in their manuscript fully available?

Reviewer #1: Yes

Reviewer #2: Yes

4. Is the manuscript presented in an intelligible fashion and written in standard English?

Reviewer #1: Yes

Reviewer #2: Yes

5. Review Comments to the Author

Reviewer #1: Congratulations for the excellent work. The authors present a very good literature revision and the aims and research questions are sound. The analytical methods are also well designed and implemented.

The authors can also discuss their results comparing with the results recently obtained for spiders in the manuscript

Reinier de Vries, et al. (2021). A small-scale analysis of elevational species richness and beta diversity patterns of arthropods in an oceanic island (Terceira, Azores). Insects, 12: 936. DOI: 10.3390/insects12100936

Reviewer #2: This paper presents novel data collected from ground dwelling arthropods and habitat type and ecological study. The question aked is certainly of relevance to the readers of such multidisciplinary journal like Plos One. Although MS show some minor grammar mistakes with some hard-readable sentences, it is written logically and represents a nice level of content. The methods are clear and robust. Basically, I have no comments regarding the statistical process as it seems that authors used them correctly with choosing suitable statistical methods. On the other hand, it would be nice to use also indicator values analysis (IndVal) to identify significant indicator species of arthropods for particular habitats along the elevation gradient. It can show which species is related to studied habitat types/environ. predictor significantly. I suggest it may bring interesting information’s for readers, which should be mentioned also in Discussion part. Authors can consider this suggestion.

Further comments:

1) Even I am not native speaker, I had feeling that English can be better in some parts of MS. I would recommend reading by native speaker which enhance the English validity of MS.

2) Line 89: (introduction): there are more suitable papers for references including importance of soil properties for ground dwelling arthropods such as spiders, millipedes, centipedes and ground beetles, also their overal potential of these groups for biodiversity analyses, see and consider adding: https://www.sciencedirect.com/science/article/pii/S030147972101803X.

3) Line 165: as you were sampling in NP, where material is stored? Maybe this can be addded to the Material and Methods section.

4) Line 243: You are using abundance level, however what about activity density? Is not it more about activity density (when you were collecting by pitfall trapping, e.g. spiders) than abundance level?

5) Line 300: Cannot be merged to one overal table??

6. PLOS authors have the option to publish the peer review history of their article (what does this mean?). If published, this will include your full peer review and any attached files.

Reviewer #1: **Yes: **Paulo A. V. Borges

Reviewer #2: No

---

## [Author Response · Author response to Decision Letter 0]

1 Jul 2022

We carefully considered the remarks and suggestions by the editor and reviewers and made the following improvements.

Editorial remark

We note that Figure 1 in your submission contain [map/satellite] images which may be copyrighted. All PLOS content is published under the Creative Commons Attribution License (CC BY 4.0), which means that the manuscript, images, and Supporting Information files will be freely available online, and any third party is permitted to access, download, copy, distribute, and use these materials in any way, even commercially, with proper attribution. For these reasons, we cannot publish previously copyrighted maps or satellite images created using proprietary data, such as Google software (Google Maps, Street View, and Earth). For more information, see our copyright guidelines: http://journals.plos.org/plosone/s/licenses-and-copyright.

Response: we added written permission by the Swiss National Park Direction as copyright holder to use the map presented in Figure 1 for this article (permit form signed by Swiss National Park director Ruedi Haller). The map presents a slightly updated version of the map in the previous version of Figure 1 with copyright data added.

As requested we also checked the style requirements and the labeling of the files and made some changes accordingly.

Reviewers' comments:

Reviewer #1: Congratulations for the excellent work. The authors present a very good literature revision and the aims and research questions are sound. The analytical methods are also well designed and implemented. 

The authors can also discuss their results comparing with the results recently obtained for spiders in the manuscript. Reinier de Vries, et al. (2021). A small-scale analysis of elevational species richness and beta diversity patterns of arthropods in an oceanic island (Terceira, Azores). Insects, 12: 936. DOI: 10.3390/insects12100936

Response: Following the suggestion of the reviewer we inserted the reference in the discussion and compared these findings with ours.

Reviewer #2: This paper presents novel data collected from ground dwelling arthropods and habitat type and ecological study. The question aked is certainly of relevance to the readers of such multidisciplinary journal like Plos One. Although MS show some minor grammar mistakes with some hard-readable sentences, it is written logically and represents a nice level of content. 

Response: The revised manuscript has been read by people with good English skills and the wording improved in some places.

The methods are clear and robust. Basically, I have no comments regarding the statistical process as it seems that authors used them correctly with choosing suitable statistical methods. On the other hand, it would be nice to use also indicator values analysis (IndVal) to identify significant indicator species of arthropods for particular habitats along the elevation gradient. It can show which species is related to studied habitat types/environ. predictor significantly. I suggest it may bring interesting information’s for readers, which should be mentioned also in Discussion part. Authors can consider this suggestion.

Response: following the suggestion by the reviewer, we calculated indicator values (IndVal) for the three vegetation types (forest, alpine grassland, patchy vegetation). However, IndVal analysis showed only some species characteristic for forest and none for grassland. We inserted these new findings as a new paragraph under the subtitle species composition. The results of the full analysis are presented in the new Table S13.

Further comments:

1) Even I am not native speaker, I had feeling that English can be better in some parts of MS. I would recommend reading by native speaker which enhance the English validity of MS.

Response: The revised manuscript has been read by people with good English skills and the wording improved in some places.

2) Line 89: (introduction): there are more suitable papers for references including importance of soil properties for ground dwelling arthropods such as spiders, millipedes, centipedes and ground beetles, also their overal potential of these groups for biodiversity analyses, see and consider adding: https://www.sciencedirect.com/science/article/pii/S030147972101803X.

Response: Following the suggestion of the reviewer we inserted the reference in the discussion.

3) Line 165: as you were sampling in NP, where material is stored? Maybe this can be addded to the Material and Methods section.

Response: As already stated in the enhanced methods given in S1 File, the majority of the collected arthropods are deposited in the Bündner Naturmuseum in Chur (Grisons). We now added this information to the methods section in the main text.

4) Line 243: You are using abundance level, however what about activity density? Is not it more about activity density (when you were collecting by pitfall trapping, e.g. spiders) than abundance level?

Response: We would agree with the reviewer if we had solely used pitfall traps to collect arthropods. However, pitfall traps were only one of three sampling methods employed. Wet cardboard sheets and visual search do not depend on the activity of the invertebrates. Therefore, the term abundance is more adequate in our case.

5) Line 300: Cannot be merged to one overal table??

Response: The raw data on the species abundances of each group (site x species matrix) is presented separately for each taxonomical group to limit the size of the table (xx species overall) and to enable easy access for readers, which will often be only interested in some of the taxonomical groups. Furthermore, our analyses were done separately for each taxonomical group and the structure of the data files in the supplementary material reflects this.

We change the corresponding authorship to Stephan Zimmermann from Brigitte Braschler.

Best regards

Stephan Zimmermann and Brigitte Braschler

---

## [Editor Report · Decision Letter 1]

8 Jul 2022

Six groups of ground-dwelling arthropods show different diversity responses along elevational gradients in the Swiss Alps

PONE-D-22-06634R1

Dear Dr. Zimmermann,

We’re pleased to inform you that your manuscript has been judged scientifically suitable for publication and will be formally accepted for publication once it meets all outstanding technical requirements.

Kind regards,

Bi-Song Yue, Ph.D

Academic Editor

PLOS ONE

---

## [Editor Report · Acceptance letter]

15 Jul 2022

PONE-D-22-06634R1 

Six groups of ground-dwelling arthropods show different diversity responses along elevational gradients in the Swiss Alps 

Dear Dr. Zimmermann:

I'm pleased to inform you that your manuscript has been deemed suitable for publication in PLOS ONE. Congratulations! Your manuscript is now with our production department. 

Kind regards, 

on behalf of

Dr. Bi-Song Yue 

Academic Editor

PLOS ONE